# Graph Neural Networks for Modeling Social Processes in Online Social Networks

## Abstract

This work considers the problem of modeling the dynamics of social activity in online communities, with graph neural networks forming the methodological basis of the approach. Observations are represented as sequences of interaction events, on the basis of which a graph structure reflecting relationships between elements of the process is constructed. A preliminary statistical analysis of time series and aggregated activity characteristics is performed to formalize the modeling task and select an appropriate feature representation. The graph neural network model is used for short-term forecasting of changes in activity levels based on early-stage discussion data, with the graph representation serving as a source of information on the influence of interaction structure on engagement dynamics. The results of the graph-based model on historical data from the Reddit social network are compared with models that use aggregated features without explicit consideration of graph structure. Preliminary results demonstrate improved forecasting performance when using graph representations, indicating the significant role of structural interaction characteristics in the formation of social process dynamics.

## 1   Introduction

Modeling the dynamics of social processes is one of the central problems of modern mathematical theory and data analysis. In online communities, social activity manifests itself as sequences of interactions — posts and comments — forming complex branching discussion structures. Such processes are characterized by nonlinear dynamics, activity bursts, and a pronounced dependence on local interactions between participants.

Several major approaches to modeling social dynamics have been developed in the literature. Continuous models, including SIR-type systems (Kermack & McKendrick, 1927) and their extensions to networked and information-spreading settings (Pastor-Satorras & Vespignani, 2001; Daley & Kendall, 1965), describe the aggregated evolution of agent populations through systems of differential equations. These approaches provide analytical interpretability, allow the study of equilibrium stability, and support parameter identification and control formulations. A systematic review of mathematical models of social processes, including inverse and control problems, is presented in (Zvonareva & Krivorotko, 2026).

Alongside continuous approaches, discrete network-based models have been developed, where dynamics are determined by the structure of interactions between agents. Classical cascade models (Kempe et al., 2003) and threshold models (Granovetter, 1978; Watts, 2002) describe the spread of influence over networks and emphasize the role of topology in shaping collective behavior. In such formulations, aggregated dynamics are directly linked to the configuration of local connections and branching interaction patterns.

Modern machine learning methods enable a transition from fixed probabilistic interaction rules to learnable interaction models. Graph Neural Networks (GNNs) implement a message-passing mechanism between neighboring vertices and allow the extraction of representations from graph-structured data. Architectures such as GCN (Kipf & Welling, 2017), Graph-

SAGE (Hamilton et al., 2017), and GAT (Veličković et al., 2018) have become widely used in network analysis tasks.

In recent years, GNNs have been actively applied to social network analysis, including influencer detection, misinformation identification (Monti et al., 2019), information diffusion modeling (Wu et al., 2020), and learning on dynamic graphs (Pareja et al., 2020). In these settings, graph-based models serve as tools for extracting structural regularities from interaction data. An overview of neural-network approaches in modeling social processes is provided in (Zvonareva & Krivorotko, 2026).

However, most existing studies focus on classification tasks or node-level influence estimation. The problem of short-term forecasting of aggregated activity dynamics based on the microstructure of discussions remains less formalized. In particular, it is unclear whether the branching structure of discussions provides additional predictive information about future activity beyond the current scale of engagement.

In other words, if the current volume of activity already serves as a strong predictor of subsequent growth, can the configuration of local interactions — discussion depth, response distribution, and reply structure — act as an early indicator of future changes in total activity?

In this work, for each calendar day, a directed interaction graph is constructed using a sliding time window. Forecasting is formulated on a fixed horizon $\Delta = 2$ days as a regression problem of relative activity change, ensuring scale invariance of the formulation. Additionally, a threshold-based formulation is considered to identify high-activity days.

The contributions of this work are as follows:

1. A formulation of short-term engagement forecasting as a graph-based regression problem with a scale-invariant target variable is proposed.

2. A sliding graph representation of discussion dynamics is developed.

3. Several graph neural network architectures (GCN, GraphSAGE, GAT) are compared in pure and hybrid regimes.

4. The contribution of structural information is quantitatively evaluated relative to models using only aggregated features.

The results demonstrate that the microstructure of discussions contains a measurable predictive signal of future activity dynamics, while its contribution is complementary to the current scale of engagement.

## 2 Data and Problem Formulation

The empirical study is conducted on discussion data related to the Yellow Vests protest movement (France, 2018–2019), obtained from public Reddit archives (subreddits r/france, r/worldnews, r/europe) for the period from October 1, 2018 to December 31, 2019.

Let $N(d)$ denote the number of comments published on day $d$. This defines a discrete time series

$$\{N(d)\}_{d=d_0}^{d_T}.$$

Forecasting is performed at a fixed horizon $\Delta = 2$ days (chosen as the minimal interval allowing detection of short-term discussion inertia without smoothing sharp activity peaks). For each day $d$, the aggregated future activity is defined as

$$S_{\text{future}}(d) = \sum_{t=d}^{d+\Delta-1} N(t),$$

which represents the total number of comments over the subsequent two days.

To analyze relative changes in activity, a historical window of length $W$ days is introduced:

$$S_{\text{past}}(d) = \sum_{t=d-W+1}^{d} N(t).$$

The target variable is defined as the logarithm of the relative change in activity:

$$y^{\text{reg}}(d) = \log \frac{S_{\text{future}}(d) + \varepsilon}{S_{\text{past}}(d) + \varepsilon}, \quad \varepsilon > 0.$$

This formulation ensures scale invariance and enables meaningful comparison across periods with different baseline activity levels.

Additionally, a binary formulation of the task is considered, where an activity surge is defined as a day whose future activity exceeds the upper quantile of the distribution of $S_{\text{future}}(d)$ on the training set:

$$\theta = Q_{0.8}\big(S_{\text{future}}(d)\big).$$

The binary label is defined as

$$y^{\text{cls}}(d) = \mathbf{1}\{S_{\text{future}}(d) > \theta\}.$$

This formulation allows evaluation of the model's ability to identify days belonging to the top 20% in terms of future activity.

## 3 Graph Representation of Discussions

For each day $d$, a directed discussion graph

$$G_d = (V_d, E_d)$$

is constructed using a sliding temporal window of length $W = 7$ days.

### 3.1 Sliding Window

Let

$$\mathcal{T}_d = \{d - W + 1, \ldots, d\}$$

denote the set of calendar days included in the historical window. The vertices of the graph are defined as the set of comments published during the period $\mathcal{T}_d$:

$$V_d = \{c : \text{comment } c \text{ is published in } \mathcal{T}_d\}.$$

Thus, each day is associated with a separate graph representing the structure of discussions over the last $W$ days. The graph is used to predict the quantity $S_{\text{future}}(d)$ associated with day $d$.

### 3.2 Graph Edges

Each comment in the dataset contains an identifier of its parent message. If comment $c_j$ is a reply to comment $c_i$ and both belong to $V_d$, then a directed edge

$$(c_i, c_j) \in E_d$$

is included in the graph.

Since Reddit discussions are organized as reply trees, the graph $G_d$ consists of a collection of directed acyclic subtrees. This results in a sparse and hierarchical structure.

### 3.3 Node Features

Each vertex $v \in V_d$ is associated with a feature vector
$$x_v \in \mathbb{R}^4,$$
including:

- the depth of the node in the discussion tree,
- $\log(1 + \text{score})$ of the comment,
- $\log(1 + \text{text length})$,
- the age of the comment (in days) relative to the end of the window.

Depth is defined as the length of the shortest path from the vertex to the root of the corresponding subtree.

### 3.4 Graph-Level Features

In addition to node-level characteristics, aggregated graph-level features are computed:
$$u_d \in \mathbb{R}^7,$$
including:

- $\log(1 + |V_d|)$ — number of vertices,
- $\log(1 + |E_d|)$ — number of edges,
- graph density,
- mean and maximum depth,
- mean comment age,
- standard deviation of comment age.

These features reflect the overall scale and structural properties of the discussion.

### 3.5 Final Representation

Thus, each day $d$ is associated with the object
$$(G_d, \{x_v\}_{v \in V_d}, u_d),$$
which serves as input to the graph neural network.

Graphs corresponding to different days are constructed independently, with no direct edges between them. Temporal dependence arises indirectly due to overlapping sliding windows, which lead to partial overlap of vertices in adjacent daily graphs.

## 4 Model

To process the graphs $G_d$, we employ graph neural networks (GNNs) that aggregate information along the edge structure and construct a fixed-dimensional graph representation. Three architectures are considered: GCN, GraphSAGE, and GAT.

### 4.1 Node-Level Message Passing

Let $x_v^{(0)} = x_v$ denote the initial feature vector of vertex $v \in V_d$. At layer $k$, the vertex representation is updated via a message passing mechanism that aggregates information from neighboring vertices:
$$x_v^{(k+1)} = \sigma \left( W^{(k)} \cdot \text{AGG} \left( x_v^{(k)}, \{x_u^{(k)} : u \in \mathcal{N}(v)\} \right) \right),$$
where $\mathcal{N}(v)$ denotes the set of neighbors of vertex $v$, AGG is an aggregation function, $W^{(k)}$ is a learnable weight matrix, and $\sigma(\cdot)$ is a nonlinear activation function (ReLU).

Two consecutive message passing layers are used in the model.

### 4.2 Considered Architectures

GCN. The GCN architecture uses normalized neighborhood aggregation, ensuring symmetric averaging of incoming messages.

GraphSAGE. GraphSAGE performs neighbor aggregation followed by a linear transformation and nonlinearity. In this work, mean aggregation is used.

GAT. GAT employs an attention mechanism to weight messages from neighboring vertices, allowing heterogeneous importance of connections.

### 4.3 Graph-Level Aggregation

After the final layer, a matrix of node embeddings is obtained:

$$H_d \in \mathbb{R}^{|V_d| \times h}.$$

A global pooling operation is applied to obtain a graph-level representation:

$$g_d = \text{POOL}(H_d),$$

where POOL denotes either mean pooling over vertices or concatenation of mean and max pooling.

### 4.4 Feature Usage Modes

Three variants of input representation are considered:

- u-only: only graph-level features $u_d$ are used;
- pure GNN: only the structural representation $g_d$ is used;
- hybrid GNN: a combined vector is used,

$$z_d = [g_d, u_d].$$

In the u-only regime, linear regression is applied to features $u_d$ without using a graph architecture. In the pure and hybrid regimes, the graph neural network described above is used.

This separation allows evaluation of the contribution of structural graph information relative to aggregated scale-based characteristics.

### 4.5 Regression Model

Based on the final representation $z_d$ (or the corresponding vector in pure/u-only regimes), a scalar output is computed:

$$\hat{y}_d = f(z_d; \theta),$$

where $f$ is a two-layer fully connected network.

In the regression task, the model approximates

$$y_d^{\text{reg}} = \log \frac{S_{\text{future}}(d) + \varepsilon}{S_{\text{past}}(d) + \varepsilon}.$$

### 4.6 Loss Function

To ensure robustness to outliers, the smoothed L1 loss (Huber loss) is used:

$$\mathcal{L}_{\text{reg}} = \frac{1}{|train|} \sum_{d \in train} \ell_\delta \left( \hat{y}_d - y_d^{\text{reg}} \right),$$

where the parameter $\delta > 0$ controls the transition between quadratic and linear regions. The function $\ell_\delta(r)$ is defined as

$$\ell_\delta(r) = \begin{cases} \frac{1}{2}r^2, & |r| \leq \delta, \\ \delta\left(|r| - \frac{1}{2}\delta\right), & |r| > \delta. \end{cases}$$

In the classification setting, the standard binary cross-entropy loss is used.

## 5 Experimental Setup and Metrics

### 5.1 Graph Construction Parameters

The length of the historical window is fixed at $W = 7$ days. The forecasting horizon is $\Delta = 2$ days.

For each day $d$, the graph is constructed using data from the period $\{d - W + 1, \ldots, d\}$. Only graphs satisfying the conditions

$$|V_d| \geq 30, \qquad |E_d| \geq 1$$

are included in the analysis.

This filtering excludes trivial or degenerate structures.

### 5.2 Data Split

A temporal split is used: the first 80% of valid days form the training set $train$, and the remaining 20% form the test set $test$.

Within the training portion, an additional validation segment is used for model selection.

### 5.3 Regression Metrics

The following metrics are used to evaluate the prediction of relative activity change.

Mean Absolute Error (MAE):

$$\text{MAE} = \frac{1}{|test|} \sum_{d \in test} |\hat{y}_d - y_d|.$$

Pearson Correlation:

$$\text{Corr} = \frac{\sum_{d \in test}(\hat{y}_d - \bar{\hat{y}})(y_d - \bar{y})}{\sqrt{\sum_d(\hat{y}_d - \bar{\hat{y}})^2}\sqrt{\sum_d(y_d - \bar{y})^2}}.$$

Coefficient of Determination:

$$R^2 = 1 - \frac{\sum_{d \in test}(\hat{y}_d - y_d)^2}{\sum_{d \in test}(y_d - \bar{y})^2}$$

Additionally, reconstruction error of the quantity $S_{\text{future}}(d)$ after inverse transformation from the logarithmic scale is analyzed.

### 5.4 Ranking Metrics

Since the test interval contains no days labeled as activity surges, standard classification metrics (ROC-AUC, PR-AUC) are not applicable on the test set.

Instead, we evaluate the model's ability to rank days by expected future activity.

Table 1: Regression results (log-ratio) on the test set

| Architecture | Mode | MAE | Corr | $R^2$ |
|---|---|---|---|---|
| u-only | – | 1.02 | 0.53 | 0.28 |
| GAT | pure | 0.87 | 0.70 | 0.45 |
| GAT | hybrid | 0.86 | 0.69 | 0.44 |
| GCN | pure | 0.85 | 0.67 | 0.42 |
| GCN | hybrid | 0.85 | 0.68 | 0.43 |
| GraphSAGE | pure | 0.86 | 0.68 | 0.42 |
| GraphSAGE | hybrid | 0.84 | 0.68 | 0.41 |

Precision@K: For a given $K$, this metric measures the fraction of overlap between the $K$ days with the highest predicted activity and the $K$ days with the highest actual activity.

Optimal Time Lag: The quantity

$$\tau^* = \arg\max_{\tau} \mathrm{Corr}(\hat{y}_{d+\tau}, y_d)$$

is computed to characterize possible anticipation of activity growth by the model.

### 5.5 Threshold-Based Surge Formulation

Within the training set, binary classification precision is additionally evaluated to analyze the model's ability to identify days belonging to the upper quantile of future activity.

## 6 Results

### 6.1 Dataset Overview

After constructing the sliding windows, a total of 455 calendar days were included in the analysis. The training set contains 364 days, and the test set contains 91 days. The surge threshold, defined as the 0.8-quantile of the distribution of $S_{\mathrm{future}}(d)$ on the training set, equals $\theta = 1300.40$. The proportion of positive labels in the training set is approximately 0.20.

The test interval contains no days labeled as activity surges ($y = 1$); therefore, standard classification metrics are not applicable on the test set. The primary focus is placed on regression and ranking metrics.

Table 1 presents the average MAE, Pearson correlation, and coefficient of determination computed over three random initializations on the test set.

The u-only regime achieves a correlation of Corr $= 0.53$, indicating the presence of predictive signal in the current activity scale.

Across all graph architectures, the pure GNN regime moderately outperforms the baseline (Corr $\approx$ 0.67–0.70), confirming the presence of additional structural information beyond aggregated scale-based features.

Differences between pure and hybrid regimes are moderate and not systematic. This suggests that the majority of structural signal is already captured through graph aggregation, while the addition of global features does not lead to substantial quality improvement.

### 6.2 Ranking Metrics

To evaluate the ability of models to rank days by future activity level, Precision@K and optimal time lag are considered.

Table 2: Ranking metrics on the test set

| Architecture | Mode | P@5 | P@10 | $\tau^*$ |
|---|---|---|---|---|
| GAT | hybrid | 0.60 | 0.50 | -2 |
| GAT | pure | 0.60 | 0.50 | -2 |
| GCN | hybrid | 0.60 | 0.50 | -2 |
| GCN | pure | 0.60 | 0.50 | -2 |
| GraphSAGE | hybrid | 0.53 | 0.53 | -2 |
| GraphSAGE | pure | 0.60 | 0.50 | -2 |

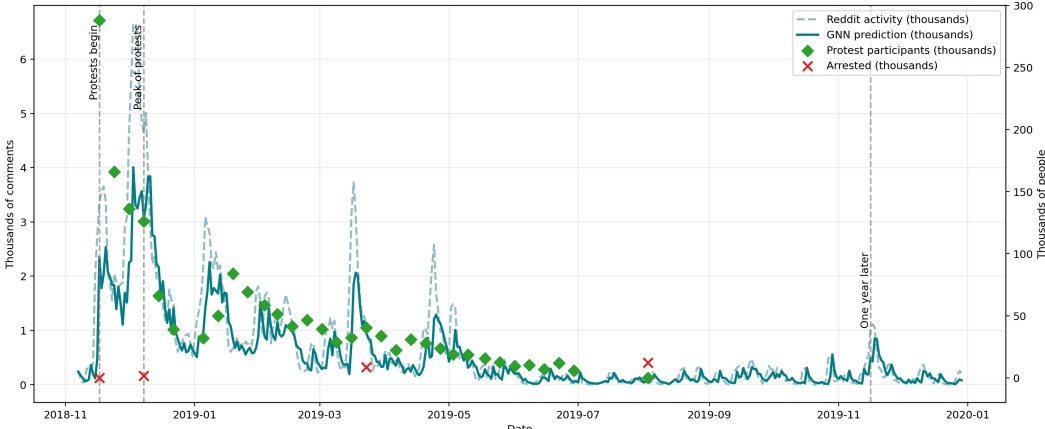

Figure 1: Qualitative comparison of temporal dynamics: online activity and offline protest indicators. The plot illustrates temporal synchronization of events.

The values of $P@K$ reflect the model's ability to identify days with the highest future growth in activity. The sign of the optimal lag $\tau^*$ characterizes the relative temporal shift between predicted and observed dynamics.

### 6.3 Comparison of Regimes

Model comparison shows:

- the linear baseline (u-only) is substantially inferior to graph-based models;

- pure GNN consistently outperforms the baseline, confirming the predictive role of discussion structure;

- the transition to hybrid mode does not yield systematic improvement.

Differences between second-order architectures (GCN, GraphSAGE, GAT) are less pronounced than the gap between linear and graph-based formulations.

## 7 Discussion

The results clarify the role of different information sources in short-term online activity forecasting.

First, the substantial improvement of graph models over the linear baseline indicates that the micro-level interaction structure contains information not reducible to aggregated discussion scale. This suggests that activity dynamics are shaped not only by overall engagement volume but also by the configuration of local interaction chains.

Second, the limited differences between second-order architectures suggest that the specific implementation of message passing is secondary in this setting. The key factor is the graph-based representation itself.

The absence of systematic improvement in the hybrid regime indicates partial correlation between global aggregated features and structural graph characteristics. This implies that a significant portion of scale-related information is implicitly encoded in the edge structure.

From a methodological perspective, the results support the use of graph-based models in tasks where the target variable emerges from local agent interactions. The magnitude of improvement depends on the forecasting horizon and the formulation of the target variable.

## 8  Conclusion

This work investigates the predictive informativeness of online discussion structure using Reddit data related to the Yellow Vests movement (2018–2019).

It is shown that graph-based representations of user interactions provide a consistent improvement over a linear model based on aggregated features. Differences between specific graph neural network architectures are secondary relative to the structural formulation of the problem itself.

The findings confirm that micro-structure of discussions contains a quantitatively measurable signal of future activity dynamics.

## Acknowledgements

This research was supported by the grant of the Federal Territory "Sirius" under the state program Scientific and Technological Development of the Federal Territory "Sirius" (Agreement No. 26-03, dated 07 July 2025).

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
