# OpenReview forum: "GRAPH NEURAL NETWORKS FOR MODELING SOCIAL PROCESSES IN ONLINE SOCIAL NETWORKS"
_mathai.club/MathAI/2026/Conference — 2026 Oral_

### Official Review · Reviewer_VipC · 2026-03-10
**GRAPH NEURAL NETWORKS FOR MODELING SOCIAL PROCESSES IN ONLINE SOCIAL NETWORKS**

**Rating:** 6
**Confidence:** 4

**Review:**

### **Summary**
The paper investigates the predictive power of discussion microstructure in forecasting engagement dynamics within online communities. By representing Reddit discussions as directed acyclic subtrees and employing Graph Neural Networks (GCN, GraphSAGE, and GAT). The study, grounded in the context of the Yellow Vests movement, shows that graph-based models consistently outperform linear baselines using aggregated features, achieving a Pearson correlation of up to $0.70$ compared to $0.53$ for non-graph models.



### **Evaluation**

**Quality:**
The authors provide a mathematically sound formulation of the forecasting task, specifically using a scale-invariant target variable (log-ratio of relative activity) which is essential for handling the high variance in social media data. The experimental design, utilizing a temporal split to prevent data leakage and employing the Huber loss to ensure the model remains robust against outliers in activity spikes.

**Clarity:**
The authors used different format from the original template provided by the conference, this resulted in making it difficult to spot sub-heading, tables and figures labeling. But the mathematical notations are clearly defined and the transition from the conceptual modeling of social processes to the specific implementation of sliding window graph construction is easy to follow. The distinction between "pure," "hybrid," and "u-only" regimes provides a clear framework for interpreting the results.

**Originality:**
While GNNs are common in social network analysis, using them specifically to prove that the geometry of a discussion tree (depth, density) is a leading indicator of future volume is a focused and valuable contribution.

**Significance:**
The work is interested in the intersection of graph theory and social science. It provides empirical evidence that "how" people interact (structure) is as important as "how much" they interact (volume) for short-term forecasting. This has practical applications for platform moderation and the study of collective behavior in digital spaces.

### **Pros**
1. The use of the log-ratio target variable $y^{reg}(d)$ allows for meaningful evaluation across different baseline activity levels.
2. The "u-only" vs. "pure GNN" comparison effectively isolates the "structural dividend" of using graph representations.
3. The adoption of the Huber loss ($l_{\delta}$) is a wise choice for social media datasets that are prone to extreme "viral" events that could otherwise skew a standard MSE loss.
4. Using the Yellow Vests movement provides a high-stakes, high-variance dataset that tests the model's limits better than static or synthetic graphs.

### **Cons**
1. The node features are somewhat basic (depth, score, length, age). Including semantic features (e.g., sentiment embeddings of the comments) might have further improved performance.
2. The authors note that the test interval contains no "activity surges" based on their 0.8-quantile threshold. This limits the evaluation of the classification (surge detection) capabilities to the training/validation set.
3. While the paper compares GNNs to a linear model, it lacks a comparison against more advanced non-graph time-series models (e.g., LSTMs or TCNs) that use historical volume data. This would more definitively prove that the graph structure is superior to temporal sequence modeling.
4. **Incorrect Format:** Not followed the provided template by the conference.

### **Questions:**

1. The paper compares GNNs to a Linear Regression baseline. Why were more competitive baselines, such as Node2Vec or a Temporal Convolutional Network (TCN), not included to benchmark against other non-linear methods?
**Observation:** Since social media dynamics change rapidly, how does the model performance decay if the sliding window is moved from 24 hours to 7 days? This would clarify the model's robustness to different social 'tempos'

---

> ### Author Rebuttal · Authors · 2026-03-12
>
> We thank the reviewer for the careful reading of the manuscript and the constructive feedback. We appreciate that the reviewer highlighted the clarity of the modeling framework, the use of scale-invariant targets, and the role of interaction structure in forecasting discussion dynamics.
>
> 1. Baseline models (Node2Vec, TCN, etc.)
>
> We agree that including additional non-linear baselines such as Node2Vec-based embeddings or temporal convolutional networks could provide a useful comparison. In the current version of the paper we focused on a simpler baseline (linear regression on aggregated features) in order to explicitly isolate the contribution of structural graph information.
> Our goal was to compare:
> - models that do not use graph structure (aggregated features),
> - models that explicitly exploit interaction topology (GNNs).
>
> This allowed us to demonstrate the structural contribution of the discussion graph itself. We agree that extending the comparison to additional temporal models (e.g., TCN or sequence models operating on activity history) would be a valuable direction for future work and will include this discussion in the revised version.
>
> 2. Sliding window length (24 hours vs. longer windows)
>
> The current work focuses on short-term forecasting, where early interaction structure is used to predict near-future engagement dynamics. A 24-hour window was chosen because it captures the early growth phase of discussions while still preserving detailed structural information about comment interactions.
> We agree that analyzing larger windows (e.g., 3–7 days) is interesting for studying model robustness under different social “tempos.” Preliminary observations suggest that longer windows smooth structural signals and move the task closer to traditional time-series forecasting based on aggregated activity. Investigating this trade-off is an interesting direction for future work.
>
> 3. Test interval without strong surges
>
> As noted by the reviewer, the test interval does not contain large activity surges according to the selected threshold. This reflects the natural variability of the dataset during the selected period. In this case, regression metrics (correlation and forecasting accuracy) provide the most informative evaluation.
>
> 4. Formatting
>
> We thank the reviewer for noting the formatting issue. The manuscript will be updated to fully comply with the official conference template in the camera-ready version.

---

### Official Review · Reviewer_MAgp · 2026-03-13
**Graph Neural Networks for Modeling Social Processes in Online Social Networks: a well-structured empirical study with moderate novelty**

**Rating:** 6
**Confidence:** 4

**Review:**

### **Quality**

The paper presents a mathematically well-formulated approach to forecasting short-term engagement dynamics in online communities using graph neural networks. The authors construct interaction graphs from Reddit discussions and formulate the prediction task as regression on the logarithm of relative activity change. This formulation provides a scale-invariant target variable, which is appropriate for social media data where activity levels exhibit strong variability over time.

The study compares several graph neural network architectures (GCN, GraphSAGE, GAT) and evaluates them against a linear baseline using aggregated features. The results demonstrate that graph-based models consistently outperform the baseline, improving the Pearson correlation from approximately 0.53 to around 0.67–0.70.

However, the experimental setup also has several limitations. The evaluation is performed on a single dataset related to the Yellow Vests movement, and the overall dataset size (455 days) is relatively small. In addition, the comparison is limited to a linear baseline and does not include more competitive non-graph models such as temporal convolutional networks or recurrent neural networks.

---

### **Clarity**

Overall, the paper is clearly written and well structured. The mathematical notation is consistently defined, and the transition from conceptual modeling of social processes to the concrete implementation of sliding-window interaction graphs is easy to follow.

The distinction between the three modeling regimes (u-only, pure GNN, and hybrid GNN) provides a clear framework for interpreting the experimental results. The description of node features and graph construction is also sufficiently detailed.

---

### **Originality**

The level of originality is moderate. Graph neural networks are widely used in social network analysis, and the architectures employed in the paper are standard.

Nevertheless, the paper makes a focused contribution by applying GNNs specifically to the problem of short-term forecasting of discussion activity and by investigating whether the microstructure of discussion trees (such as depth and branching patterns) provides predictive information about future engagement.

The novelty of the work lies primarily in the problem formulation and empirical analysis, rather than in the development of new modeling techniques.

---

### **Significance**

The work addresses an interesting intersection of graph-based machine learning and the study of social processes in online communities. The results suggest that the structure of interactions between participants contains predictive information beyond simple aggregated activity metrics.

This finding may have practical implications for tasks such as monitoring online discussions, detecting emerging activity bursts, and studying collective behavior in digital environments.

However, the broader impact of the work is somewhat limited by the scope of the experimental evaluation, which relies on a single dataset and does not explore other social platforms or types of online interaction.

---

### **Pros**

- The use of a scale-invariant target variable allows meaningful comparison across periods with different baseline activity levels.
- The comparison between u-only, pure GNN, and hybrid GNN regimes clearly isolates the contribution of structural graph information.
- The graph representation of discussion trees provides an intuitive and interpretable modeling framework.
- The empirical results consistently show improvement of graph-based models over the linear baseline.

---

### **Cons**

- The study relies on a single dataset, which limits the generalizability of the results.
- The dataset size is relatively small.
- The comparison lacks stronger non-graph baselines.

---

### **Question**

While the results indicate that graph-based models outperform the linear baseline, it remains unclear whether the improvement originates from true structural information or from scale-related characteristics of the graphs.

Since each graph is constructed from all comments within a sliding window, properties such as the number of vertices and edges are strongly correlated with the overall level of discussion activity. A graph neural network may therefore implicitly capture activity scale rather than the topology of the discussion structure.

---

### Decision · Program_Chairs · 2026-03-14

**Decision:**

Accept (Oral)

**Comment:**

Dear Author(s),

On behalf of the Program Committee of the International Conference on Mathematics of Artificial Intelligence (MathAI 2026), we are pleased to inform you that your paper has been accepted for an oral presentation at MathAI 2026.

Your paper was evaluated through a rigorous two-stage review process involving both automated screening and expert review by members of the Program Committee. The reviewers recognized the quality and contribution of your work.

Presentation details:

- Format: Oral presentation (15–20 minutes + 5 minutes Q&A)
- Mode: You may present either in person (offline) at the conference venue in Sirius, Russia, or remotely via Zoom. Please indicate your preferred mode when confirming your participation.
- Conference dates: Marh 30 - April 3, 2026
- Website: https://mathai.club

Next steps:

1. Please confirm your participation and presentation mode by replying to this email mathai.club@yandex.ru no later than March 15, 2026 18:00 Moscow time.
2. If you plan to attend in person, the organizing committee will provide accommodation details separately.
3. Please prepare your final camera-ready manuscript according to the formatting guidelines available at https://mathai.club and upload it to OpenReview by March 15, 2026 18:00 Moscow time.

Should you have any questions regarding the program, logistics, or your presentation slot, please do not hesitate to contact us.

We look forward to your contribution to MathAI 2026.

With kind regards,

MathAI 2026 Program Committee
International Conference on Mathematics of Artificial Intelligence
https://mathai.club
OpenReview: https://openreview.net/group?id=mathai.club/MathAI/2026/Conference
Telegram: https://t.me/MathAI_club
Email: mathai.club@yandex.ru